# Assortative mating biases marker-based heritability estimators

Richard Border [1,2,3 ✉], Sean O'Rourke[4], Teresa de Candia[2], Michael E. Goddard[5,6], Peter M. Visscher [7], Loic Yengo[7], Matt Jones[8,9] & Matthew C. Keller [2,8,9 ✉]

Many traits are subject to assortative mating, with recent molecular genetic findings confirming longstanding theoretical predictions that assortative mating induces long range dependence across causal variants. However, all marker-based heritability estimators implicitly assume mating is random. We provide mathematical and simulation-based evidence demonstrating that both method-of-moments and likelihood-based estimators are biased in the presence of assortative mating and derive corrected heritability estimators for traits subject to assortment. Finally, we demonstrate that the empirical patterns of estimates across methods and sample sizes for real traits subject to assortative mating are congruent with expected assortative mating-induced biases. For example, marker-based heritability estimates for height are 14% – 23% higher than corrected estimates using UK Biobank data.

[1] Departments of Neurology and Computer Science, University of California, Los Angeles, California, USA. [2] Institute for Behavioral Genetics, University of Colorado Boulder, Colorado, USA. [3] Department of Epidemiology, Harvard T.H. Chan School of Public Health, Massachusetts, USA. [4] Department of Mathematics, University of Colorado Boulder, Colorado, USA. [5] Faculty of Veterinary and Agricultural Science, University of Melbourne, Victoria, Australia. [6] Department of Economic Development, Jobs, Transport and Resources, Biosciences Research Division, Victoria, Australia. [7] Institute for Molecular Bioscience, University of Queensland, Saint Lucia, QLD, Australia. [8] Department of Psychology and Neuroscience, University of Colorado Boulder, Colorado, USA. [9]These authors contributed equally: Matt Jones, Matthew C. Keller. ✉email: border.richard@gmail.com; matthew.c.keller@gmail.com

Positive primary phenotypic assortative mating (hereafter simply "AM"), the phenomenon whereby mate-choice is based on phenotypic similarity, has been observed for a variety of heritable traits in human and non-human animals[1–4]. A century ago, Fisher demonstrated that AM induces long-range positive correlations between trait-increasing allele counts at causal loci across the genome, thereby increasing genetic variance across successive generations until it approaches a stable equilibrium[5]. Since Fisher's time, it has been established that many human traits are subject to AM, and that estimates of genetic and environmental variance from twin and family designs, which assume random mating, can be biased in the presence of AM[6]. Recently, there has been increased focus on how AM can influence results based on measured genetic data, in part due to the recognition that AM may contribute to observed discrepancies between population-based versus family-based estimates in genome-wide association studies (GWAS)[7–10] and Mendelian randomization studies[11,12]. However, despite hundreds of publications reporting estimates of SNP-heritability, no study has yet investigated how AM influences these estimates. Moreover, many benchmark traits central to the scientific discourse regarding the "still-missing heritability"—for example, height and educational attainment—are precisely those for which phenotypic and genetic data is consistent with AM[4,13–15], raising the possibility that conclusions about SNP-heritability could be biased in systematic but as yet poorly understood ways.

Here, we characterize the impact of AM on the two families of marker-based heritability estimators, which we generically denote $\hat{h}_{\mathrm{SNP}}^2$. The first uses the method of moments (MoM) and is typified by univariate Haseman-Elston (HE) regression ($\hat{h}_{\mathrm{HE}}^2$)[16] but includes PCGC regression[17] and linkage disequilibrium (LD) score regression[18]. The second uses residual maximum likelihood (REML)[19] to estimate heritability ($\hat{h}_{\mathrm{REML}}^2$) and is implemented in software such as by GCTA[20] and BOLT-REML[21]. We assume Fisher's classical model of AM, which describes the equilibrium properties of a heritable trait for which mates' genotypes are conditionally independent given the heritable components of their phenotypes and for which offspring environments are independent of parental phenotypes. This model has formed the theoretical foundation for recent investigations of AM using measured genetic data[4,14,22]. We provide mathematical and simulation-based arguments demonstrating that AM induces a modest yet non-negligible bias in both classes of estimators that is not addressed by conventional methods of accounting for population structure. In the process, we extend results in random matrix theory and classical quantitative genetics by characterizing the higher-order moments of causal variants and the limiting spectral distribution of the genomic relatedness matrix (GRM) under AM, which clarifies why genomic principal components fail to capture the effects of AM. Additionally, we provide empirical results using data from the UK Biobank that are congruent with our theoretical predictions. Finally, we provide guidelines for estimating and interpreting $\hat{h}_{\mathrm{SNP}}^2$ when traits are subject to AM.

## Results

Our theoretical results depend on several key parameters: $r$ denotes the phenotypic correlation between mates on an additive phenotype $\mathbf{y}$ with a heritable component $\mathbf{Zu}$; $h_0^2$ denotes the panmictic heritability, what the narrow-sense heritability of the phenotype would be in the absence of AM; $h_\infty^2$ denotes the equilibrium narrow-sense heritability approached under multiple generations of AM; and $\mathbf{Z}$ denotes the $n \times m$ matrix of $n$ unrelated individuals' standardized genotypes at $m$ causal loci with effects vector $\mathbf{u}$. We initially assume that all causal variants are present in

$\mathbf{Z}$ (i.e., that the heritability is fully explained by measured variants); the reason for this is that the dynamics of AM across generations depend on the true heritability of the trait, irrespective of the fraction of genetic variation tagged by SNPs. We later employ simulations to demonstrate that our major qualitative conclusions remain valid when this assumption is relaxed.

The rows of $\mathbf{Z}$ (individuals' genotypes) are independent random vectors with $m \times m$ covariance matrix $\Upsilon$, which quantifies the correlation between loci. Under random mating, $\Upsilon := \Upsilon_0$ is approximately block diagonal such that causal variants are largely (aside from LD between nearby variants) stochastically independent. However, under equilibrium AM, $\Upsilon := \Upsilon_\infty$ is dense due to the presence of positive long-range correlations among trait-increasing allele counts within and across chromosomes. As the elements of $\Upsilon_\infty$ agree in sign with the corresponding elements of $\mathbf{uu}^\mathrm{T}$ (i.e., trait increasing alleles are positively correlated), the equilibrium additive genetic variance under AM, $\sigma_{g,\infty}^2$, is considerably greater than the panmictic additive genetic variance, $\sigma_{g,0}^2$. That is, $\sigma_{g,\infty}^2 = \mathbf{u}^\mathrm{T}\Upsilon_\infty\mathbf{u} > \sigma_{g,0}^2 \approx \mathbf{u}^\mathrm{T}\mathbf{u}$ (Fig. 1 and Supplementary Notes 1, 2).

**Haseman–Elston regression estimates under AM.** We first characterize the influence of AM on $\hat{h}_{\mathrm{HE}}^2$, perhaps the simplest marker-based MoM heritability estimator. Let $\widetilde{\mathbf{y}}$ denote the standardized phenotype. $\hat{h}_{\mathrm{HE}}^2$ is the slope of the subdiagonal elements of the phenotypic outer product, $\widetilde{\mathbf{y}}\widetilde{\mathbf{y}}^\mathrm{T}$, regressed on the subdiagonal elements of the GRM, $m^{-1}\mathbf{Z}\mathbf{Z}^\mathrm{T}$. Assuming that all genetic variance is explained by measured variants, we establish the following general result (see Supplementary Note 3):

$$\mathbb{E}\left[\hat{h}_{\mathrm{HE}}^2\right] = \left(\frac{m \cdot \mathbf{u}^\mathrm{T}\Upsilon_\infty\Upsilon_\infty\mathbf{u}}{\sigma_{g,\infty}^2 \cdot \mathrm{tr}\left[\Upsilon_\infty\Upsilon_\infty\right]}\right)h_\infty^2 \geq h_\infty^2 \geq h_0^2. \quad (1)$$

Under AM, the bracketed quantity is greater than one, and thus $\hat{h}_{\mathrm{HE}}^2$ is upwardly biased relative to both $h_0^2$ and $h_\infty^2$. Intuitively, this bias occurs because AM influences the outcome ($\widetilde{\mathbf{y}}\widetilde{\mathbf{y}}^\mathrm{T}$) differently than the predictor ($m^{-1}\mathbf{Z}\mathbf{Z}^\mathrm{T}$). The AM-induced positive correlations between all causal variants inflate genetic (co-)variance, and this inflation is accurately reflected in $\widetilde{\mathbf{y}}\widetilde{\mathbf{y}}^\mathrm{T}$. However, the elements of $m^{-1}\mathbf{Z}\mathbf{Z}^\mathrm{T}$ represent the average correlation between pairs of individuals at homologous loci taken one at a time. Thus, the elements of $m^{-1}\mathbf{Z}\mathbf{Z}^\mathrm{T}$ are governed almost completely by the correlations within loci; correlations between different loci have almost no influence. Because the vast majority of the inflation in genetic (co-)variation is caused by the correlations between and not within loci, AM on phenotype has a negligible impact on the GRM for polygenic traits in a genetically homogenous population. This latter point is itself noteworthy because some studies have claimed that inflated genomic similarity between friends or mates[22] is evidence for phenotypic assortment, but the actual increase is trivial (of order $O(m^{-1})$ relative to the increase in genetic variance) and such observations have more probable explanations, such as imperfectly controlled stratification[23].

Under the stricter assumption of exchangeable loci (i.e., each causal variant explains equal phenotypic variance), we derive the following approximate expression, dependent only on $r$ and $h_0^2$ (see Supplementary Note S3):

$$\mathbb{E}\left[\hat{h}_{\mathrm{HE}}^2\right] \approx \left(\frac{1}{1-rh_\infty^2}\right)h_\infty^2. \quad (2)$$

Under exchangeable loci for known $r$ and $\hat{h}_{\mathrm{HE}}^2$, we further define corrected estimators of the panmictic and equilibrium

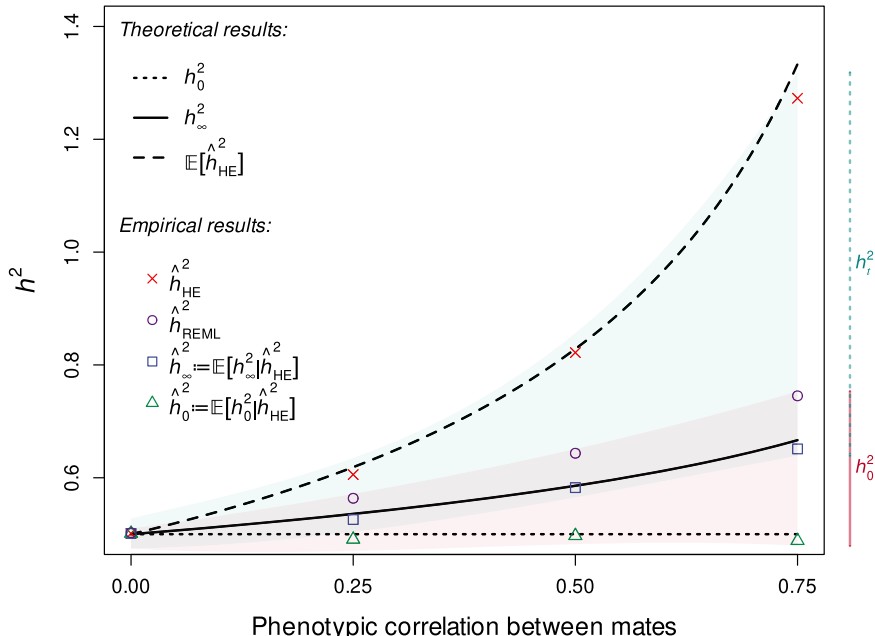

**Fig. 1 Theoretical and empirical behavior of existing and corrected estimators.** HE regression ($\hat{h}^2_{HE}$) and REML ($\hat{h}^2_{REML}$) estimates for varying phenotypic mate correlations ($r$) in simulated datasets ($n = 64,000$) assuming that $h^2_0 = .5$, that all causal variants are measured, and that AM has reached equilibrium. Values of $\hat{h}^2_{HE}$ are consistent with our closed-form approximation ($\mathbb{E}[\hat{h}^2_{HE}]$) under the assumption of exchangeable loci. Our corrected estimators, $\hat{h}^2$ and $\hat{h}^2_0$, which are based on $\hat{h}^2_{HE}$ and $r$ and assume equilibrium, recover the true equilibrium ($h^2_\infty$) and panmictic ($h^2_0$) heritabilities. Further, for a given observation of $\hat{h}^2_{HE}$ for a trait under disequilibrium AM, the current generation heritability ($h^2_t$) and $h^2_0$ are probabilistically bounded in expectation (teal and red regions respectively; see main text).

heritability values:

$$\hat{h}^2_0 := \mathbb{E}\left[h^2_0 | \hat{h}^2_{HE}\right] = \frac{\hat{h}^2_{HE}}{1 + 2r\hat{h}^2_{HE} + r(r-1)\hat{h}^4_{HE}},$$

$$\hat{h}^2_\infty := \mathbb{E}\left[h^2_\infty | \hat{h}^2_{HE}\right] = \frac{\hat{h}^2_{HE}}{1 + r\hat{h}^2_{HE}}. \tag{3}$$

Large-scale simulations using realistic genotype data (see Online Methods) with $p = 10^6$ SNPs of which $m = 10^4$ were causal with effects $u \overset{i.i.d.}{\sim} \mathcal{N}(0, \sigma^2_{g,0}/m)$ demonstrate that these approximations are accurate even when the exchangeable loci assumption is violated (Figs. 1,2). Results were nearly identical when setting $m = 10^5$ causal variants. In addition, our simulations confirmed that LDSC, which is mathematically equivalent to HE regression when LD scores are exact[24], is similarly biased upwards under AM (Fig. 3a). However, the impact of this bias in real world applications depends not only on the extent of AM, but also on the degree to which estimated LD scores reflect the true LD structure in a given population.

**Residual maximum likelihood estimates under AM.** In contrast to $\hat{h}^2_{HE}$, the value of $\hat{h}^2_{REML}$ changes as a function of sample size under AM. Using the same simulation approach described above, we found that $\hat{h}^2_{REML}$ exhibits upward biases similar in magnitude to those of $\hat{h}^2_{HE}$ in small samples (e.g., $n \approx 10,000$). In large samples (e.g., $n > 200,000$), however, $\hat{h}^2_{REML}$ drops below $h^2_\infty$ (Fig. 2a). More formally, under the assumption of exchangeable loci, we prove that

$$\hat{h}^2_{REML} \overset{p}{\to} h^2_0, \quad \text{as } n, m \to \infty, \tag{4}$$

where $n/m \to c \in (0, \infty)$; i.e., $\hat{h}^2_{REML}$ is a consistent estimator of

$h^2_0$ for highly polygenic traits (Supplementary Note 3). On the other hand, the number of causal variants and the total number of measured SNPs have no apparent influence on the bias of $\hat{h}^2_{REML}$ (Supplementary Fig. 1). In essence, the parameter values that maximize the residual likelihood function depend only on the eigenvalues of the GRM, and the long-range dependence among causal variants induced by AM is "weak" in the sense that the distributions of the eigenvalues (i.e., spectral distributions) of the GRM under random mating and under AM are asymptotically equivalent. Thus, in large samples, $\hat{h}^2_{REML}$ converges to what the heritability would be if the causal variants were independent, i.e., to $h^2_0$. However, in finite samples, we show via simulation that this convergence can be extremely gradual, requiring samples approaching millions of individuals before $\hat{h}^2_{REML}$ approaches $h^2_0$ (Fig. 2b).

**Conventional methods do not mitigate AM-induced bias.** Inclusion of ancestral PCs as covariates fails to mitigate the AM-induced bias in both the MoM and the REML estimates (Fig. 3a). Indeed, we demonstrate that AM has a negligible effect on the spectral distribution of the GRM in large samples with many variants (see Supplementary Note 2). Similarly, these biases are not mitigated by modeling multiple genetic variance components that partition SNPs according to LD score and minor allele frequency (Fig. 3a).

**AM-induced bias persists when not all causal variants are measured.** In real-world applications, measured genotypes will often include only a fraction of the total number of causal variants (or at least good proxies in high LD). Previous work has demonstrated that the extent to which marker data imperfectly tag causal variants will bias $\hat{h}^2_{REML}$ estimates in random mating populations[25,26]; these circumstances are further complicated under AM as imperfect tagging will not only corrupt the

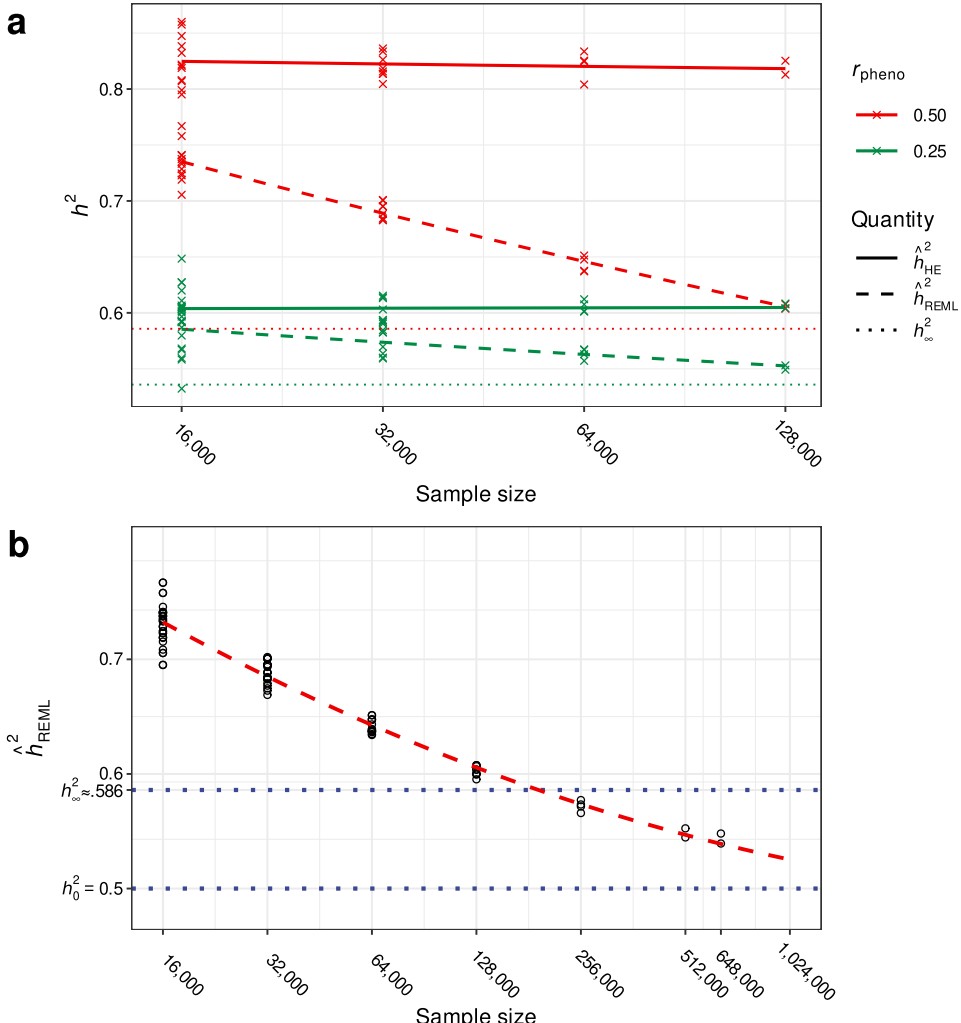

**Fig. 2 REML and HE estimates across varying sample sizes in simulated data. a** Comparison of HE regression and REML heritability estimates as functions of sample size for varying phenotypic mating correlation ($r_{pheno}$) and fixed panmictic heritability ($h_0^2 = 0.5$) in simulated data. We computed multiple estimates per sample size for each estimator and parameter combination by applying estimators to independent sub-samples. Whereas HE regression estimates are upwardly biased independent of sample size, REML estimates slowly converge to the panmictic heritability as sample sizes increase. **b** Extended simulations demonstrating high-dimensional behavior of the REML estimator as a function of sample size. Forward time simulations required a larger population size ($N_{sim} = 3 \times 10^6$) to obtain samples of up to $n = 648,000$ unrelated individuals. Obtaining REML estimates for samples larger than this was not computationally feasible, but the dashed red line shows predicted values for larger sample sizes extrapolated from a regression model including first and second order log-linear components. Results are consistent with theoretical predictions that the REML estimator converges to the panmictic heritability in very large samples.

relationship between a given marker and proximal causal variants, but between a given marker and all other causal variants. To assess the impact of AM when not all variants are measured, we compared $\hat{h}_{HE}^2$ and $\hat{h}_{REML}^2$ in simulated data that randomly dropped $\pi = 0$, 50, or 75% of all $p = 10^6$ variants, including both causal ($m = 10^4$) and non-causal ($p - m = 990,000$) SNPs (Fig. 3b). As expected, this resulted in attenuated estimates commensurate with, but not proportional to the fraction of missing data: on average across methods, $\hat{h}_{SNP}^2$ was 8.4% (s.e. 0.009%) and 17.6% (s.e. 0.009%) lower after randomly dropping 50% and 75% of SNPs, respectively (the degree of attenuation was smaller than the proportion of SNPs dropped due to LD between retained SNPs and discarded causal variants). Though future work is required to fully characterize the relationship between heritability estimates and the degree of missing heritability under AM, the pattern of bias due to AM when some of the heritability is missing is qualitatively similar to that when all causal variants are present.

**The differential influence of sample size on $\hat{h}_{HE}^2$ and $\hat{h}_{REML}^2$ in real traits**. To investigate whether our theoretical predictions are observable in real data, we examined the relationship between sample size and $\hat{h}_{SNP}^2$ in a sample of 335,551 unrelated European-ancestry individuals in the UK Biobank[27]. We a priori selected four phenotypes based on evidence (height, years of education) or lack of evidence (body mass index, bone mineral density) for primary phenotypic AM in a previous study[14]. We then computed $\hat{h}_{HE}^2$ and $\hat{h}_{REML}^2$ in pairs of small ($n_{small} = 16,000$) versus large ($n_{large} = n - 16,000$) disjoint subsamples, where $n$ denotes the total available sample size for each phenotype. Consistent with theory and simulation results, the effect of sample size on $\hat{h}_{REML}^2$ was significantly different from the effect of sample size on $\hat{h}_{HE}^2$ for height ($p = 5.24e{-}4$) and for years of education ($p = 3.94e{-}4$); this was not observed for body mass index ($p = 0.094$) or for bone mineral density ($p = 0.302$; see "Methods" section, Fig. 4). These

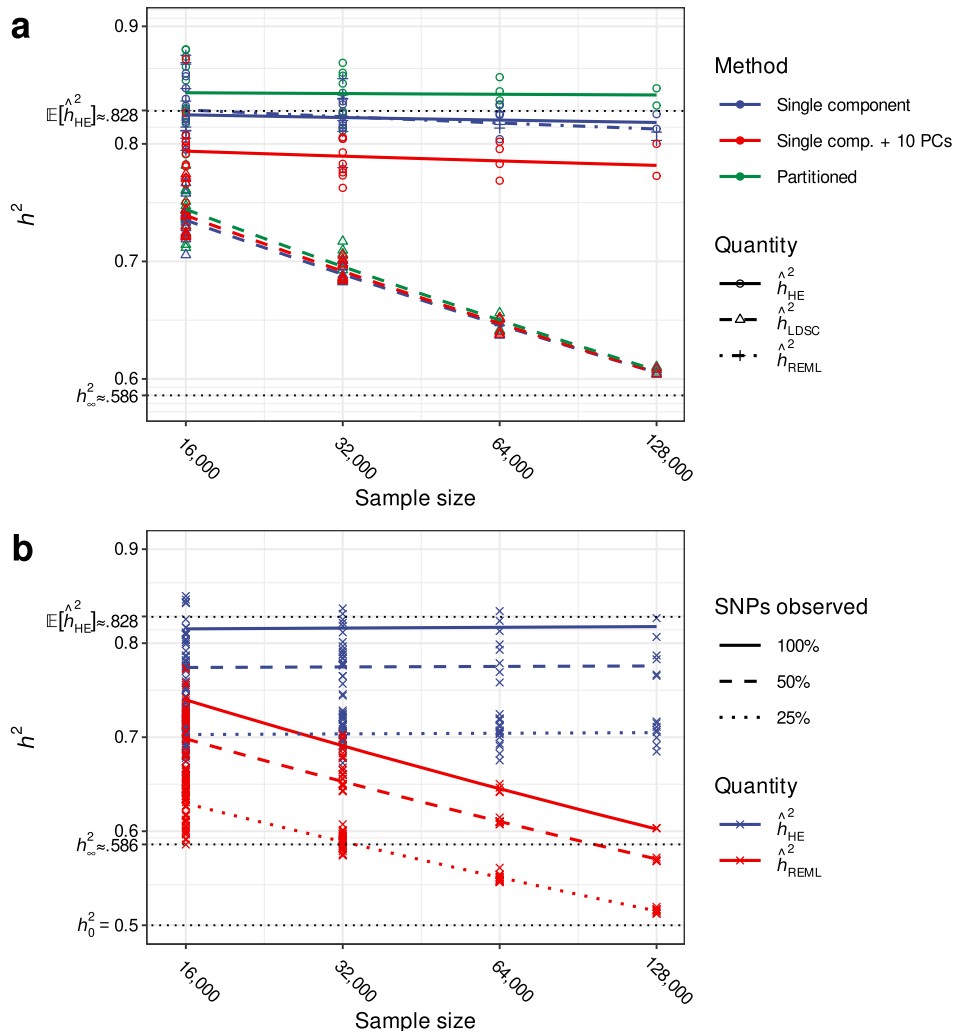

**Fig. 3 Naïve approaches to addressing AM induced bias and the impact of missing data. a** Simulations employing the same parameters described in Fig. 1 demonstrate that neither partitioned nor principal component adjusted approaches mitigate the impact of AM on HE ($\hat{h}^2_{HE}$) or REML ($\hat{h}^2_{REML}$) heritability estimates. Additionally, simulations confirm that LDSC is subject to equivalent biases. Single component: standard single genomic variance component models. Single comp. + 10 PCs: included the first ten within-sample PCs as covariates. Partitioned: included four annotation-based variance components generated by median splits of within-sample minor allele frequencies and LD scores. **b** Simulations demonstrate that conclusions regarding estimator bias do not change when some of the influence of causal variants is not captured by measured SNPs. Simulations employed the same parameters as above except that 0, 50, or 75% of randomly selected SNPs (both causal and non-causal) were dropped. As expected, estimates were attenuated when SNPs were dropped but overall patterns remained consistent.

height results are congruent with those of Mandal and colleagues, who independently observed that $\hat{h}^2_{REML}$ decreased with increasing sample size[28]. Unlike previous approaches which quantified the degree of AM for traits by correlating polygenic scores between mates[4] or within-individuals but between chromosomes[14], our approach does not require the calculation of polygenic scores and is agnostic with respect to variant direction and effect size. Thus, calculating $\hat{h}^2_{REML}$ across subsamples of varying sizes provides an alternative and independent way to detect genomic signatures of AM.

**Interpretation of $\hat{h}^2_{SNP}$ in the presence of AM.** As we have shown, AM complicates the interpretation of $\hat{h}^2_{SNP}$. This complication is due not only to the differential effect AM has on $\hat{h}^2_{HE}$ and $\hat{h}^2_{REML}$ as a function of sample size, but also because AM changes the true heritability in the population. This means that $\hat{h}^2_{SNP}$ can be compared to two different population parameters: the

equilibrium heritability $h^2_\infty$, and the panmictic heritability $h^2_0$. We input $\hat{h}^2_{HE}$ and previously reported spousal correlations for height[2] and educational attainment[4] into Eq. (3) to estimate $h^2_\infty$ and $h^2_0$. $\hat{h}^2_{HE}$ was inflated by 14% for height and 7% for years of education relative to $\hat{h}^2_\infty$ and by 21 and 14% relative to $\hat{h}^2_0$ (Table 1). We note that this underestimates the absolute bias of $\hat{h}^2_{HE}$ with respect to $h^2_\infty$ and $h^2_0$ to the extent that causal variants are missing from our measured genotype data.

Although these estimates of $h^2_\infty$ and $h^2_0$ rely on the assumption that AM has reached equilibrium, theoretically-sound bounds for $h^2$ can be derived for traits where AM is at disequilibrium. Specifically, the current heritability for a population having undergone $t$ generations of AM, $h^2_t$, is bounded in expectation from above by $\hat{h}^2_{HE}$, with equality at $t = 0$ (panmixis), and from below by $\hat{h}^2_\infty = \mathbb{E}\left[h^2_\infty | \hat{h}^2_{HE}; t = \infty\right]$, with equality as $t \to \infty$ (equilibrium). Likewise, $h^2_0$ is bounded in expectation from below

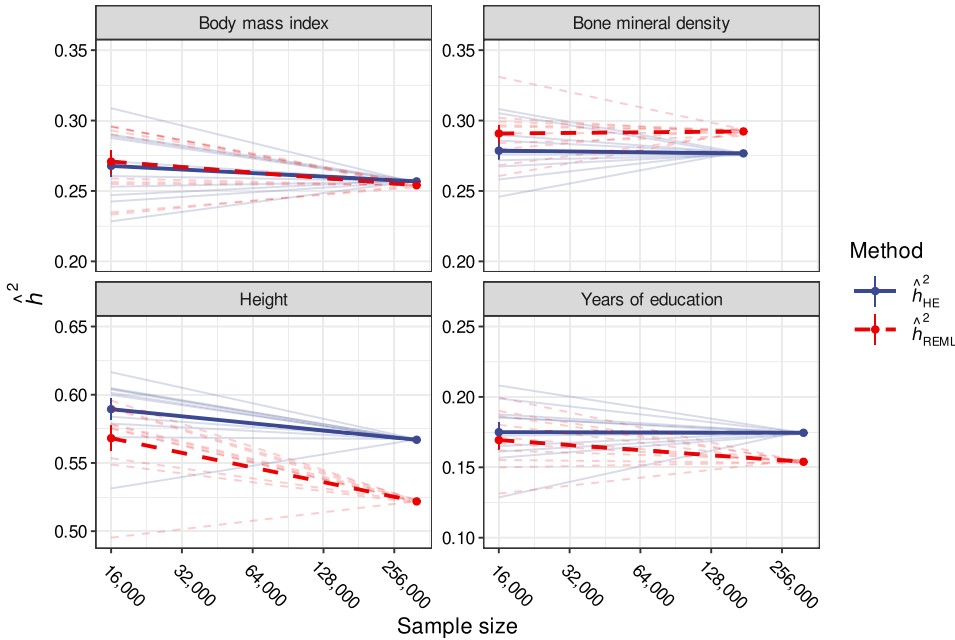

**Fig. 4 REML and HE estimates across varying sample sizes in UK Biobank data.** Comparison of HE ($\hat{h}^2_{HE}$) and REML ($\hat{h}^2_{REML}$) heritability estimates as a function of sample size for real traits in a sample of unrelated European ancestry UK Biobank participants. Points connected by thin lines represent estimates derived from pairs of non-overlapping subsamples of size 16,000 and $n-16{,}000$, whereas thick lines reflect average log-linear trends. Two traits with previous evidence for AM (height and years of education) and two negative control traits (body mass index and bone mineral density) were selected for analysis a priori. Consistent with theoretical predictions, height and years of education demonstrated significant estimator divergence with increasing sample size whereas body mass index and bone mineral density did not.

**Table 1 Inflation of heritability estimates and their corrected values for select UK Biobank traits.**

| Phenotype | r | n | $\hat{h}^2_{HE}$ [s.e.] | $\hat{h}^2_{REML}$ [s.e.] | $\hat{h}^2$ [s.e.] | $\hat{h}^2_0$ [s.e.] | $\hat{h}^2_{HE}/\hat{h}^2$ [s.e.] |
|---|---|---|---|---|---|---|---|
| Body mass index[a] | 0.228 | 334,429 | 0.257 [3.15e−3] | 0.256 [2.29e−3] | – | – | – |
| Bone mineral density[a,b] | – | 191,330 | 0.277 [4.37e−3] | 0.295 [3.28e−3] | – | – | – |
| Height | 0.240 | 334,798 | 0.567 [4.68e−3] | 0.525 [2.22e−3] | 0.499 [4.23e−3] | 0.467 [3.37e−3] | 1.14 [8.70e−4] |
| Years of education | 0.412 | 332,198 | 0.174 [2.64e−3] | 0.155 [2.01e−3] | 0.163 [2.46e−3] | 0.153 [2.06e−3] | 1.07 [9.46e−4] |

Spousal correlations (r) as previously reported in British cohorts[2,4] and heritability estimates for selected UK Biobank phenotypes (n denotes sample size). Height and years of education were selected a priori on the basis of previous evidence for primary phenotypic AM whereas body mass index and bone mineral density were selected a priori as negative controls. Assuming equilibrium, the equilibrium ($\hat{h}^2$) and panmictic ($\hat{h}^2_0$) heritability estimates (defined in Eq. 3) provide unbiased estimates of the present day and panmictic heritabilities, respectively. Under disequilibrium, they respectively provide probabilistic lower bounds, with the HE regression ($\hat{h}^2_{HE}$) and REML ($\hat{h}^2_{REML}$) estimates providing complementary upper bounds. The ratio $\hat{h}^2_{HE}/\hat{h}^2$ reflects the extent to which HE regression overestimates the true heritability under the assumption of equilibrium.
[a]Adjusted equilibrium and panmictic heritability estimates omitted for negative control traits.
[b]No estimates of the phenotypic mating correlation for bone mineral density were available.

by $\hat{h}^2_0 = \mathbb{E}\left[h^2_0 | \hat{h}^2_{HE}; t=0\right]$ with equality as $t \to \infty$, and from above by $\hat{h}^2_{REML}$ with equality at all generations. This latter bound will narrow as sample size increases. Thus, regardless of whether AM is at equilibrium, so long as the strength of AM has not decreased across generations, $[\hat{h}^2_\infty, \hat{h}^2_{HE}]$ provides probabilistic bounds for the present day $h^2$ whereas $[\hat{h}^2_0, \hat{h}^2_{REML}]$ provides probabilistic bounds for $h^2_0$ (Fig. 1).

## Discussion

**Summary of findings**. In the last decade, there has been much interest in the interpretation of $\hat{h}^2_{SNP}$ and the factors that can influence it[29]. Such factors include the number of causal variants and their typical levels of LD compared to the LD of SNPs used in the analysis[30,31], the relationship between causal variant effect sizes and minor allele frequencies[32], and passive gene–environment correlation arising from population stratification[33] or genetic

nurture[7]. Despite this activity and the recent evidence corroborating long-standing theoretical expectations that AM alters the genetic architecture of heritable traits[14], the effects of AM on $\hat{h}^2_{SNP}$ have remained a matter of speculation[9,30,34]. In the present investigation, we clarify how AM influences $\hat{h}^2_{SNP}$ and show that these influences are different for MoM versus REML estimators as a function of sample size. By characterizing the full joint distribution of causal variants at equilibrium, we prove that REML produces a consistent estimator of $h^2_0$ (the heritability in the ancestral random-mating population), not $h^2_\infty$ (the present generation heritability for a trait at equilibrium), in large samples (see Supplementary Notes 2 and 3). However, under AM, $\hat{h}^2_{REML}$ behaves oddly in finite samples. It is higher than $h^2_\infty$ in sample sizes typical of those published in the literature but drops below $h^2_\infty$ in sample sizes that are large by current standards (e.g., $n > 200{,}000$). On the other hand, MoM estimators yield estimates that are biased upwards with respect to both $h^2_\infty$ and $h^2_0$ and remain stable across sample sizes. Using UK

Biobank data, we observed this differential behavior of $\hat{h}_{HE}^2$ and $\hat{h}_{REML}^2$ for two traits with previous evidence of primary phenotypic AM but not for two negative control traits. We further showed how the two population parameters of interest, $h_\infty^2$ and $h_0^2$, can be estimated under the assumption of equilibrium given spousal correlations and $\hat{h}_{HE}^2$, and that, at disequilibrium, the likely ranges of $h^2$ in the current generation and $h_0^2$ can also be estimated.

**Implications**. The impact that AM has on $\hat{h}_{SNP}^2$ may often be negligible simply because AM is very low or nonexistent for many traits. For other traits, however, this does not appear to be the case, including "benchmark" traits like height and educational attainment as well as many behavioral and psychiatric traits. For example, the tetrachoric correlations between Swedish spouses for attention deficit hyperactive disorder ($r = 0.31$), autism spectrum disorder ($r = 0.28$), schizophrenia ($r = 0.26$), and substance use disorder ($r = 0.30$) are all higher than those typically observed for height[35]. Furthermore, many social attitudes such as conservatism-liberalism ($r = 0.61$) and religiosity ($r = 0.71$) exhibit extremely high spousal correlations, and available evidence is more consistent with a primary phenotypic model than social homogamy or convergence[36]. Such social attitude traits appear to be moderately heritable ($\hat{h}^2 \approx 0.20–0.30$) in extended twin family designs robust to biases caused by AM[37], although no values of $\hat{h}_{SNP}^2$ for such social attitude traits are yet available.

Our findings suggest caution is warranted when comparing $\hat{h}_{SNP}^2$ for traits subject to AM when characteristics of the populations, samples, or designs differ. First, the effect that AM has on causal variant correlations is rapid—about half of its effect occurs after a single generation and it approaches equilibrium within 5–15 generations; as such, both the true heritability and the extent to which $\hat{h}_{SNP}^2$ is biased will differ across populations that vary in strength or duration of AM. Second, because the expectation of $\hat{h}_{REML}^2$ decreases monotonically in $n$, comparing $\hat{h}_{REML}^2$ across samples of different sizes can create artifactual discrepancies. Third, though both $\hat{h}_{REML}^2$ and $\hat{h}_{HE}^2$ are expected to increase as the fraction of causal variants included in the model increases, the relative bias of $\hat{h}_{REML}^2$ will further increase if the inclusion of additional markers comes at the cost of decreased sample size. Therefore, the difference in REML estimates from a smaller sample of whole-genome sequence data (which ostensibly includes previously missing causal variants) versus those derived from a larger sample of SNP array data will appear artificially greater, though we note that additional factors, such as differences in the extents to which array versus sequence data are corrupted by subtle forms of population structure, might also contribute to such discrepancies. Fourth, our results demonstrate that the controlling for genomic principal components does not mitigate AM-induced biases; indeed, the asymptotic distributions of the eigenvalues of the GRM under random versus assortative mating are equivalent. As AM is in essence a form of population structure, our results reveal that our current understanding of the extent to which residual population structure might impact estimators of quantitative genetic parameters is incomplete. Finally, our results suggest that AM complicates comparisons of $\hat{h}_{SNP}^2$ to estimates of heritability derived from family-based methods (mostly from twins, hence $\hat{h}_{twin}^2$). On one hand, as we have shown, AM reduces the gap between $\hat{h}_{twin}^2$ and $\hat{h}_{SNP}^2$ by biasing $\hat{h}_{SNP}^2$ up. However, AM also influences $\hat{h}_{twin}^2$ depending on the design, biasing it downwards in "ACE" models that estimate additive genetic and shared environmental variance

and upwards in "ADE" models that estimate additive genetic and dominance genetic variance. Together with the previously discussed dependence of $\hat{h}_{SNP}^2$ on method and sample size, this further problematizes efforts to quantify the "still-missing" heritability ($\hat{h}_{twin}^2 - \hat{h}_{SNP}^2$).

**Limitations and future directions**. There are several limitations of the current approach. First among these are assumptions inherent to the primary phenotypic AM model. Some of these assumptions, including equilibrium and constancy of the phenotypic mating correlations across generations, aid with mathematical tractability but are to some extent inessential to the resulting phenomena. For example, while the problem of characterizing the joint distribution of causal variants is more difficult in a population subject to disequilibrium AM, we observed that estimators behave in a similar, albeit less extreme, fashion relative to their behavior in an equilibrium population. Other assumptions, such as the absence of gene–environment correlation (which may occur, for example, in the presence of vertical transmission) and the conditional independence of mates' genotypes given the heritable components of their phenotypes (which may be violated in structured populations), are more difficult to evaluate and deserve consideration in future investigations. For instance, when environmental influences are transmitted from parents to offspring, the dependence between causal variants will extend to non-heritable factors such that, in subsequent generations, all trait-increasing allele-counts and trait-increasing environmental factors will become positively correlated, and the linear mixed model assumed by both the REML and MoM estimators will be further misspecified.

We have also not investigated the impact that AM has on polygenic scores or within-family classes of estimators. For example, LD-score regression can be estimated based upon regression estimates from within-sibling analyses. The estimated genetic variance from such an analysis would be that in the base population (because it is based on the segregation variance), but the estimated phenotypic variance would be that in in the current population. This would also lead to a bias in heritability estimation (and similarly for heritability estimates using relatedness disequilibrium regression[10]), as was pointed out in ref. 38. This bias is different than those described here in that it is lower than either the base population or equilibrium population heritability. Additionally, we have not found closed-form solutions to $\hat{h}_{REML}^2$ that would allow us to correct REML estimates in the same way we have corrected HE estimates under the assumption of equilibrium AM. Further, we have not investigated the impact that natural selection has on the long-range correlations between causal variants and SNPs and what impact this might have on $\hat{h}_{SNP}^2$. Finally, we have not considered the effects of multivariate AM or the impact of AM on marker-based genetic correlation estimators, which will also be misspecified under cross-trait AM. As such, these results should provide motivation and a starting place for the development of new methods that can provide unbiased estimates of genomic variance in the presence of AM and other factors that can influence the long-range correlations between SNPs.

## Methods
### Theoretical framework
*The primary phenotypic assortment model.* Here we introduce the model of AM as proposed by Fisher[5] and further developed by Nagylaki and others[39,40] (see Supplementary Note 1 for a detailed exposition). Briefly, we consider a phenotype as a

random vector composed of independent heritable and non-heritable components:

$$\mathbf{y} = \mathbf{Z}\mathbf{u} + \mathbf{e}, \quad \mathbf{e} \overset{i.i.d.}{\sim} \mathcal{N}(0, \sigma_e^2), \tag{5}$$

where the rows of $\mathbf{Z}$, representing individuals' standardized genotypes, are independent $m$-dimensional random vectors following a multivariate discrete distribution with finite moments and finite support and which we assume are independent under panmixis. The vector of allele substitution effects $\mathbf{u}$, which we treat as fixed, is such that $\mathbf{u}^T\mathbf{u} = \sigma_{g,0}^2$. Further, we assume that 1. parent–parent–offspring trios' phenotypes are jointly Gaussian; 2. the phenotypic correlation between mates, $r$, is constant across generations; and 3. there exists $c_0 \in (0, \infty)$ such that $\max_{k=1,\dots,m} |u_k| \le c_0 \cdot m^{-1/2}$; that is, as traits become increasingly polygenic, the maximal variance attributable to individual variants decreases commensurately.

*The equilibrium distribution of causal variants.* Over successive generations, the correlation between mates' phenotypes induces positive correlations across trait increasing allele counts independent of physical position on the genome and thereby increases the total genetic variance of the trait. The genetic variance rapidly approaches a stable equilibrium after several generations (typically within ten generations), at which point the within-individual and cross-mate correlations among causal variants are equal to one another. Using the results of Nagylaki[39], we can express the equilibrium covariance matrix between causal variants as a low rank perturbation of a diagonal matrix of the form: $\Upsilon_\infty = \mathbf{D} + 2\phi\phi^T$, where $\phi$ is a known vector-valued function of the substitution effects and mating correlation (see Supplementary Note 1.2) with elements $\phi_k = O(m^{-1/2})$ uniformly.

*Higher order moments and the limiting spectral distribution of GRM.* Employing tools from the study of thermodynamic equilibria, we extend these classical results to bound moments of higher orders (see Supplementary Note 2). Using these results, we extend the widely-known Marčenko-Pastur theorem, which describes the limiting distribution of the spectrum of sample covariance matrices corresponding to random matrices with independent sub-Gaussian elements[41], to the case of random matrices with independent rows meeting particular moment conditions (see Supplementary Note 2). Together, these results establish the limiting spectral distribution of the sample GRM (i.e., the distribution of the eigenvalues of the sample GRM as both sample size and polygenicity increase) under AM, providing the necessary theoretical foundation to characterize the asymptotic behavior of the REML estimator. Further, these results explain why controlling for principal components fails to remove AM-induced biases: the impact of AM on the spectrum of the GRM is asymptotically negligible.

*Haseman-Elston regression under AM.* The HE regression estimator[42] is obtained by regressing the subdiagonal elements of the standardized phenotypic outer product $\widetilde{\mathbf{y}}\widetilde{\mathbf{y}}^T$ on the subdiagonal elements of the GRM $m^{-1}\mathbf{Z}\mathbf{Z}^T$. Whereas elements of the outcome (the phenotypic outer product) reflect the dependences among all pairs of causal loci:

$$\mathbb{E}\left[\{\widetilde{\mathbf{y}}\widetilde{\mathbf{y}}^T\}_{i,j<1}\right] \propto \sum_{k=1}^{m}\sum_{l=1}^{m} u_k u_l \mathbb{E}\left[z_{ik} z_{jl}\right], \tag{6}$$

elements of the GRM only capture the dependences within loci:

$$\mathbb{E}\left[\{\mathbf{Z}\mathbf{Z}^T\}_{i,j<1}\right] \propto \sum_{k=l}^{m} \mathbb{E}\left[z_{ik} z_{jl}\right] \tag{7}$$

As a result, the variance of the outcome increases whereas the variance of predictor remains largely unaffected, leading to overestimation of $h_{SNP}^2$ (Fig. 1; see Supplementary Note 3 for further details and proof).

*REML and the spectrum of the GRM under AM.* The REML estimator[19] models the phenotype as a random vector with marginal distribution,

$$\mathbf{y} \sim \mathcal{MVN}\left(\mathbf{X}\boldsymbol{\beta}, m^{-1}\mathbf{Z}\mathbf{Z}^T\sigma_g^2 + \mathbf{I}\sigma_e^2\right), \tag{8}$$

where $\mathbf{X}$ is an $n \times c$ matrix of covariates with fixed effects $\boldsymbol{\beta}$ and the covariance structure is comprised of a heritable component ($\sigma_g^2$ times the GRM) and a non-heritable component ($\sigma_e^2$ times the identity). The heritability estimator $\hat{h}_{REML}^2 = \hat{\sigma}_g^2/(\hat{\sigma}_g^2 + \hat{\sigma}_e^2)$ is derived by finding the values of the variance components that satisfy the equation,

$$\nabla \ell\left(\hat{\sigma}_g^2, \hat{\sigma}_e^2 | \mathbf{A}^T\mathbf{y}\right) = 0, \tag{9}$$

where $\ell$ denotes the marginal log likelihood of the transformed random variable $\mathbf{A}^T\mathbf{y}$ for $\mathbf{A}^T : \mathbb{R}^n \to (\text{col } \mathbf{X})^\perp \subseteq \mathbb{R}^{n-c}$, $\mathbf{A}^T\mathbf{A} = \mathbf{I}$. The conditional expectation of $\nabla \ell$ given the genotypes is a function of the eigenvalues of the GRM and, as a result, the asymptotic behavior of $\hat{h}_{REML}^2$ is governed by the asymptotic distribution of the eigenvalues of $m^{-1}\mathbf{Z}\mathbf{Z}^T$. A foundational result in random matrix theory states that for zero-mean unit-variance sub-Gaussian random matrices $\mathbf{W} \in \mathbb{C}^{n \times m}$ with independent elements, the empirical spectral distribution function of $m^{-1}\mathbf{W}\mathbf{W}^T$

converges almost surely to the Marčenko-Pastur distribution[41]. Employing this result, Jiang and colleagues[43] demonstrated that, in the case of independent causal variants, REML consistently estimates the true heritability in high dimensional settings and is robust to certain forms of model misspecification. In Supplementary Note 2, we demonstrate that even though AM induces dependence among causal variants, this dependence is "weak" in the sense that it doesn't change the limiting spectral distribution of the GRM, thereby allowing us to apply arguments in line with those of Jiang and colleagues' (see Supplementary Note 3). Intuitively our result can be summarized as follows: as the sample size and the number of causal variants become large, the eigenvalues of the GRM under AM behave as if the causal variants were independent (as is largely the case under random mating). The behavior of the REML estimator is determined by the behavior of the eigenvalues of the GRM, and thus $\hat{h}_{REML}^2$ converges to what the heritability would be if the causal variants were independent, i.e., the panmictic heritability.

**Simulation studies.** We employed a realistic forward-time simulation framework to generate genotypic and phenotypic data. We then used these data to motivate and verify theoretical results. Below, we describe the general framework and specific simulations we performed.

*Simulation framework.* Given a recombination map and $n_{input}$ individuals' phased biallelic genotypes at $p$ diploid loci as input, we divided the genome into $k \ll p$ contiguous, non-overlapping 50kB intervals to obtain a collection of blocks, the disjoint union of which comprises the genome. Recombination events, which occurred with probabilities dictated by the recombination map, were restricted to interval boundaries, thus dramatically reducing the number of haplotypes that to keep track of while maintaining high genomic resolution and realistic LD. To achieve a target population size $N_{sim} > n_{input}$, $N_{sim}$ pairs of the $n_{input}$ individuals were non-monogamously 'mated' (i.e., matched and subject to meiosis), resulting in a new generation of $N_{sim}$ individuals whose genomes could be represented in terms of the $n_{input}$ haplotype blocks they inherited at each of the $k$ intervals. We then repeated this random mating procedure for an additional five generations, resulting in $N_{sim}$ chimeric combinations of the original $n_{input}$ genotypes while maintaining the LD structure of the original data. These chimeric genotypes comprised the input for the principal AM simulations.

At the beginning of each particular AM simulation with prespecified panmictic heritability $h_0^2 = \sigma_{g,0}^2/(\sigma_{g,0}^2 + \sigma_e^2)$, phenotypic correlation between mates $r$, $p$ SNPs, and $m$ diploid causal loci $z_1, \dots, z_m$, $m \ll p$, the standardized allele substitution effects $u_1, \dots, u_m$ were independently drawn from a Gaussian distribution with expectation zero and variance $\sigma_{g,0}^2/m$. Unless otherwise stated, all simulations used $p = 10^6$ SNPs. At each generation, phenotypes were constructed via $\mathbf{y} = \mathbf{Z}\mathbf{u} + \mathbf{e}$ where $\mathbf{e}$ was i.i.d. Gaussian with zero expectation and variance $\sigma_e^2$. Next, mates were matched according to their respective phenotypes $y_i$, $y_j$ such that $\text{corr}(y_i, y_j) \approx r$[44]. This was achieved by drawing $N_{sim}$ independent doubles $\{(\mathbf{w}^*, \mathbf{w}^{**})^T\}_{k=1}^{N_{sim}} \sim \mathcal{N}\left(\begin{bmatrix} 0 \\ 0 \end{bmatrix}, \begin{bmatrix} 1 & r \\ r & 1 \end{bmatrix}\right)$ from which $N_{sim}$ pairs of indices $\{(i,j)\}_{k=1}^{N_{sim}}$ were constructed such that $(i,j)_k$ were the positions of $w_k^*$ and $w_k^{**}$ after concatenating and sorting each element of $\{(\mathbf{w}^*, \mathbf{w}^{**})^T\}_{k=1}^{N_{sim}}$. Similarly, $N_{sim}$ indices $l = l_1, \dots, l_{N_{sim}}$ were constructed such that $l_k$ indexed the $k$th largest of the $N_{sim}$ simulated phenotypes. Finally, each $k$th mating pair was determined by taking the $l_{\lfloor i_k/2 \rfloor}$th and $l_{\lfloor i_j/2 \rfloor}$th replicates. Having chosen mates, meiosis occurred as previously detailed to construct the next generations' genotypes. All simulations were conducted in Python v3.6.8[45] using the numpy v1.16.2[46], scipy v1.2.1[47], and dask v1.1.4[48] libraries.

*Simulations using UK Biobank data.* For each simulation, the input data were derived from phased, imputed genotypes at $p = 10^6$ randomly selected imputed SNP loci in a sub-sample of $n_{input} = 435,301$ European UK Biobank participants[27]. Retained SNPs met the following criteria: minor allele frequency >0.01, Hardy–Weinberg $p$-value >$10^{-6}$, INFO score >0.95, and presence on the 1000 Genomes Phase 3 (1KG3) reference panel[49]. Genotype data were then phased to the 1KG3 reference panel in batches of 40,000 individuals using Eagle v2.4[50]. Data were then grown to a population of $N_{sim} = 10^6$ chimeric genotypes and subjected to an additional five generations of random mating as described above before being subject to AM.

We conducted AM simulations for varying mating correlations, $r \in \{0, .25, .5, .75\}$ and numbers of causal variants, $m \in \{10^4, 10^5\}$, with panmictic heritability fixed at $h_0^2 = .5$. Each simulation consisted of fifteen generations of AM and produced results congruent with classical theory. Prior to heritability estimation, close relatives $\hat{\pi} \ge .05$, were removed using GCTA v1.93.1[20], resulting in an average sample size of 141,667 across simulated datasets. Additionally, we ran a limited number of larger, more computationally intensive simulations ($N_{sim} = 3 \times 10^6$) with mating correlations fixed at $r = .5$ to investigate the large sample behavior of the REML estimator, resulting in at least 648,000 unrelated individuals across simulated datasets. There were no apparent differences across simulations as

a function of $m$, $p$, or $N_{sim}$ (Supplementary Fig. 1). We note that these simulations assume a homogenous population without any population structure apart from that induced by primary phenotypic AM; thus, the irrelevance of $p$ here does not necessarily speak to the consequences of increasing the number of variants under consideration by decreasing allele frequency or imputation quality score thresholds, as is common practice, if the additional variants capture additional fine-scale population structure.

*Heritability estimation in simulated data.* We split each simulated genotype–phenotype dataset into collections of random subsamples mutually exclusive within collection but not across collections, yielding 16 samples of 16,000 individuals, eight samples of 16,000 individuals, four samples of 32,000 individuals, two samples of 64,000 individuals, and one sample of 128,000 individuals. We then performed HE regression and single-component REML for each subsample (Fig. 2a). We used GCTA v1.91.3b[20] to construct genomic related matrices and perform HE regression. We obtained $\hat{h}^2_{REML}$ using BOLT-LMM v2.3.4[51] for computational efficiency; though BOLT-LMM uses a randomized algorithm, its numerical accuracy is comparable to that of the exact algorithm implemented GCTA[52].

We also performed a variety of supplementary analyses for a limited set of simulation parameters ($r = .5$, $h^2_0 = .5$, and $m \in \{10^4, 10^5\}$, $N = 10^6$). To demonstrate that including genomic PCs as covariates does not mitigate the impact of AM, we included ten PCs as covariates in the HE regression and REML analyses. For the former, HE regression was conducted in LDAK v5.0[53], as the HE regression implementation in GCTA cannot accommodate covariates. To demonstrate that the behavior of LDSC under AM is equivalent to that of HE regression (assuming that the LD scores accurately reflect the LD structure of the sample), we used PLINK v1.9[54] to obtain GWAS summary statistics and LDSC v1.0.1[18] to estimate within-sample LD scores using a one centiMorgan sliding window and to perform LDSC regression (Fig. 4a). To demonstrate that multiple variance component (also known as partitioned approaches[30,55]) do not mitigate the impact of AM, we fit multicomponent HE regression and REML after partitioning SNPs by minor allele frequency and LD score (Fig. 4a). Finally, to assess the scenario wherein a nontrivial fraction of causal variants are missing from $\mathbf{Z}$, we estimated HE regression and REML models after removing 50 or 75% of simulated SNPs at random (Fig. 4b).

### Statistics

*Sampling procedures.* We analyzed 1,211,273 biallelic 1KG3 SNPs with in-sample minor allele frequency >0.01, Hardy–Weinberg $p$-value >$10^{-6}$, and INFO scores >0.95, in a sample of 335,551 unrelated European UK Biobank participants[27]. We selected phenotypes a priori on the basis of previous evidence for AM; we chose height ($n = 334,798$) and years of education ($n = 332,198$) as traits with previous evidence of primary phenotypic AM, whereas we chose body mass index ($n = 334,429$) and bone mineral density ($n = 191,330$) as negative control traits[14]. We measured years of education following the procedures detailed in ref. [9]. For the results reported in Table 1, we used previously reported estimates of phenotypic spousal correlations[2,4]. For height specifically, we combined the estimates from multiple British cohorts included in a meta-analysis[2] via inverse variance weighting.

*Analysis/resampling.* We tested for evidence of AM by comparing $\hat{h}^2_{HE}$ and $\hat{h}^2_{REML}$ in small and large samples. We randomly selected ten mutually exclusive subsamples of $n_{small} = 16,000$ individuals for each trait and compared HE and REML estimates in each subsample to the non-overlapping complementary subsample comprised of the remaining $n_{large} = n - 16,000$ individuals, controlling for sex, age, genotyping batch, testing center, and the first ten genomic ancestry PCs. To eliminate variance in heritability estimates due to chance differences in covariate effect estimates across subsamples, we adjusted genotypes and phenotypes in the full sample prior to all following analyses. To our knowledge, existing software is incapable of efficient REML analysis using adjusted genotypes (analogous to dosages) in large samples; e.g., BOLT-REML requires hard-calls as input, whereas GCTA and LDAK have cubic complexity in the number of individuals and markers and would require weeks to run on a high thread-count server. We therefore utilized a modified Python implementation of the REML algorithm presented in ref. [52] (available at https://github.com/rborder/SL_REML[56]). We used LDAK v5.0 to obtain adjusted HE regression estimates[53]. In order to quantify the divergence of $\hat{h}^2_{HE}$ and $\hat{h}^2_{REML}$ as a function of $n$, we performed the following test, analogous to a $t$-test of the interaction effect in a 2 × 2 within-subjects experimental design:

$$H_0 : \bar{\delta} := (\hat{h}^2_{REML}(S) - \hat{h}^2_{REML}(S^C)) - (\hat{h}^2_{HE}(S) - \hat{h}^2_{HE}(S^C)) = 0,$$
$$H_1 : \bar{\delta} \neq 0,$$

where $S$ denotes a given large subsample and $S^c$ its complementary small subsample. Though this procedure accounts for the dependence among estimates derived in the same subsamples, the individual observations were derived from various partitionings of the same data and do not constitute truly independent observations. Therefore, the $p$-values for this test reported in the main text should be interpreted as descriptive despite our application of inferential procedures.

All statistical analyses and visualizations were conducted in R v3.5.0[57] using the ggplot2 v3.3.3[58] and MASS v7.3.49[59] libraries.

**Ethical approval.** Ethics approval for the UK Biobank study was obtained by the UK Biobank team from the North West Center for Research Ethics Committee (11/NW/0382). Access to the UK Biobank data was granted to principal investigator Dr. Matthew C. Keller (researcher ID 16651).

**Reporting summary.** Further information on research design is available in the Nature Research Reporting Summary linked to this article.

## Data availability

Data are available through the UK Biobank Access Management System (https://www.ukbiobank.ac.uk/enable-your-research/apply-for-access). A full catalogue of the data examined, including raw materials and descriptive statistics, is available via the online showcase (https://biobank.ndph.ox.ac.uk/ukb). A full description of the study protocols has been provided by the UK Biobank team[60].

## Code availability

The REML implementation utilized is available online (https://github.com/rborder/SL_REML[56]).

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

## Acknowledgements

R.B. was supported by the National Institutes of Health (T32NS048004). R.B., P.M.V., L.Y., M.J., and M.C.K. were supported by the National Institutes of Health (MH100141). L.Y. and P.M.V. were supported by the Australian Research Council (DE200100425, FL180100072). S.O. was supported in part by the National Science Foundation (DMS-1810500).

## Author contributions

T.d.C., M.E.G., P.M.V., and M.C.K. conceived of and initiated the research. R.B. performed all simulations and empirical data analyses. R.B., S.O., and M.J. derived the mathematical results. R.B. and M.C.K. drafted the primary manuscript. R.B., P.M.V., L.Y., M.J., and M.C.K. contributed to the interpretation and presentation of results, the generation of follow-up hypotheses, and the editing process.

## Competing interests

The authors declare no competing interests.
