## [Peer Review File · Nature Communications]

Assortative Mating Biases Marker-based Heritability EstimatorsREVIEWER COMMENTS

Reviewer #1 (Remarks to the Author):

This is a well-written paper describing the potential biases induced by assortative mating in heritability estimated obtained from unrelated samples. The paper is comprehensive, and the examples are highly relevant. The authors proposed an elegant correction to estimates obtained from an extended version of the Haseman-Elston approach, and simulation results from a more broadly used REML approach to estimate heritability. The paper is of general interest to the Nature Communication audience and has provides insights on the estimate of “unexplained” heritability that has been the focus of interest in many investigations. Although the investigations rely on many assumptions that are unlikely to be met in real applications (i.e. all SNPs have the same effect on the trait), the conclusions appear to be robust to some of these assumptions and provide useful guidelines for future investigations.

Here are some comments that the authors should address.

Page 4, line 14: “AM has a negligible impact on the GRM for polygenic traits.” This statement is not entirely accurate, as the GRM will be impacted by AM with respect to ancestry. This is acknowledged by the author on line 18 of page 4 (“... such as imperfectly controlled stratification.”). The observation that AM with respect to ancestry will inflate estimates of GRM was reported by Sebro et al. in PLOS Genetic 2017 (Structured mating: patterns and implications). The authors should comment on this issue of inflated GRM when there is some ancestry-related correlation among mates, and how this might affect heritability estimates and conclusions from this paper.

Authors should consider adding bone mineral density and BMI to Table 1 to provide a contrast with traits with lower spousal correlation.

Online method: assumption that E is i.i.d: E may not be i.i.d. due to the effect of other traits influencing the trait of interest (e.g. weight influencing the risk of disease). What are the possible consequences of violation of this assumption? Authors should comment.

Reviewer #3 (Remarks to the Author):

I think this is a timely and important paper and definitely needs to be published. The authors have done such a thorough job that I don't have much to complain about. If I could say anything, I would only say that it would be good if they could give simple explanations for why h_0 differs from h_{inf} and why REML asymptotes to h_{inf} and not h_0 in the main text (as opposed to the methods and Sup Mat).

The fact that REML converges to h_0 is interesting because it makes the inherent assumption that the effects are independent, yet independence is not necessary for it to be applicable as what James Lee and I tried to show (Lee and Chow. “Conditions for the Validity of SNP-Based Heritability Estimation.” Human Genetics 133, no. 8 (2014): 1011–22. <https://doi.org/10.1007/s00439-014-1441-5>). I wonder if the new machinery developed in the Sup Mat could elucidate those results further.

Reviewer #4 (Remarks to the Author):

Dear editor and authors:

The study by Border et al. addresses important issues related to SNP-heritability estimates, including both the effect of assortative mating (AM, the main focus), and finite-sample biases (and the consistency of) two commonly used estimation methods,

The paper is well written and represents an important contribution. I hope my comments will help the

authors to further improve their manuscript.

Major comments

1) Variance components versus 'heritability' (I am referring to parameters, not estimates). The derivations presented by the authors can be used to trace the effects of AM on the variance of genetic values in a population. However, establishing the consequences of AM on heritability is more difficult because AM will also induce a correlation of environmental effects (habits) that will also be passed to progeny, thus inducing a genetic-by-environment correlation. This issue is mentioned (but somehow tangentially) in the discussion. I think that this needs to be further highlighted in the manuscript (including possible abstract and introduction) because this is a very serious concern. (I am personally more interested in variances than variance-ratios!)

2) Should we care about panmictic or real-population heritability? I would argue that for most applications we are interested in the proportion of variance of phenotypes explained by genetic values in the population, and not on that proportion in (a conceptual) panmictic population. This is addressed in the study, which derives bias relative to h_{∞}^2 and h_0^2 . Perhaps this distinction may need to be further strengthened in the discussion.

Note: In the manuscript, you present a result showing that the REML estimate converges to the heritability in the panmictic population; however, this assumes exchangeability which may not hold under some circumstances (e.g., selection).

3) The demonstration that REML estimates can have a sizable finite sample bias is extremely significant. The same finding was reported four years ago by Mandal et al. (JASA, 2017, see link below) using data from UK-Biobank and human height.

4) BIAS of REML estimates. When the amount of information about variance parameters is limited, the REML functions are typically flat around the maximum (which leads to SE) and, importantly, asymmetric (with a long right tail) which sets the conditions for the estimator to be upwardly biased. As the sample size increases, the likelihood becomes sharper and increasingly symmetric, and the bias decreases. In trying to dig into possible factors that may affect bias, I have observed that the shape of the likelihood (or REML) function depends on the % of variance explained by SNPs, sample size, and the number of SNPs. You state that the bias of REML estimates is not affected by the number of SNPs. I am not convinced that this is the case. I've been seriously concerned about the upward bias of estimates derived with ultra-high-density SNPs (e.g., imputed genotypes). Many papers using ultra-high-density SNPs claim that the missing heritability is no longer there; I have not been convinced that is the case.

Other important comments

5) The fact that LD contributes to variance and that this is not fully accounted for in traditional models (which assume iid SNP effects) have been discussed in de los Campos et al. (PLoS Genetics, 2015). I believe that reference is highly relevant for your work.

6) Your work may also have implications for estimates of genetic (or SNP-correlations) as imperfect LD between markers and QTL, and LD between loci can have consequences on covariances as well. the consequences of LD (and imperfect LD) on genetic (and SNP-based) covariances are more difficult to predict than the same consequences on variances (e.g., Gianola et al., GENETICS, 2018). What would happen for instance when both traits are subject of AM in both (or opposite) directions? These issues may be worth discussed as well.

7) "Finally, we demonstrate that these biases affect real traits subject to assortment." I don't think the empirical results demonstrate the biases (because the true parameter values are unknown). Consider re-writing.

8) The conclusions derived about biases when not all causal variants are genotyped (page 6) are not very convincing to me. We see a drop in SNP-heritability estimates, but that does not lead to concluding that biases are equally relevant when causal variants are genotyped compared with an imperfect LD situation. Consider re-writing or elaborating a bit more.

Mandal et al (2017):

<https://www2.amstat.org/meetings/jsm/2017/onlineprogram/AbstractDetails.cfm?abstractid=324734>

Reviewer #1 (Remarks to the Author):

This is a well-written paper describing the potential biases induced by assortative mating in heritability estimated obtained from unrelated samples. The paper is comprehensive, and the examples are highly relevant. The authors proposed an elegant correction to estimates obtained from an extended version of the Haseman-Elston approach, and simulation results from a more broadly used REML approach to estimate heritability. The paper is of general interest to the Nature Communication audience and has provides insights on the estimate of “unexplained” heritability that has been the focus of interest in many investigations. Although the investigations rely on many assumptions that are unlikely to be met in real applications (i.e. all SNPs have the same effect on the trait), the conclusions appear to be robust to some of these assumptions and provide useful guidelines for future investigations.

Here are some comments that the authors should address.

1. Page 4, line 14: “AM has a negligible impact on the GRM for polygenic traits.” This statement is not entirely accurate, as the GRM will be impacted by AM with respect to ancestry. This is acknowledged by the author on line 18 of page 4 (“... such as imperfectly controlled stratification.”). The observation that AM with respect to ancestry will inflate estimates of GRM was reported by Sebro et al. in PLOS Genetic 2017 (Structured mating: patterns and implications). The authors should comment on this issue of inflated GRM when there is some ancestry-related correlation among mates, and how this might affect heritability estimates and conclusions from this paper.

This is a fair point and we agree our original language was too broad. We have now amended this statement to make it clear that we are referring to assortment directly on phenotypes in the context of homogenous population, writing

Because the vast majority of the inflation in genetic (co-)variation is caused by the correlations between and not within loci, AM on phenotype has a negligible impact on the GRM for polygenic traits in a genetically homogenous population

With respect to the impact of ancestry, the conclusions in the current manuscript are unaltered provided that the confounding effects of ancestry are addressed by controlling for principal components. As both MoM and REML estimators operate on genotypes after projecting out principal components, the inflation described by Sebro et al. will not affect our conclusions. On the other hand, inclusion of principal components does not prevent the AM induced biases we describe. We emphasize this on page 6, writing

Inclusion of ancestral PCs as covariates fails to mitigate the AM-induced bias in both the MoM and the REML estimates (Figure 4a). Indeed, we demonstrate that the AM has a negligible effect on the spectral distribution of the GRM in large samples with many variants (Supp. Materials S2.4).

2. Authors should consider adding bone mineral density and BMI to Table 1 to provide a contrast with traits with lower spousal correlation.

We have added these traits to Table 1 as requested.

3. Online method: assumption that E is i.i.d: E may not be i.i.d. due to the effect of other traits influencing the trait of interest (e.g. weight influencing the risk of disease). What are the possible consequences of violation of this assumption? Authors should comment.

The assumption here is only that E is independent of an individual's genotype. The "i.i.d." notation in section 1.1 of the supplement was actually superfluous and hence misleading. We have thus removed and further clarify

...where E represents the non-heritable component of Y and is assumed to be independent of the genotypes.

A related assumption we make is that offspring values of E are independent of parental values (i.e., there is no vertical transmission). When this assumption is violated, AM will induce correlations not only among causal variants but between causal variants and environmental factors. We now note this assumption and suggest this as a topic for future work in the Discussion section, writing

There are several limitations of the current approach. First among these are assumptions inherent to the primary phenotypic AM model. Some of these assumptions, including equilibrium and constancy of the phenotypic mating correlations across generations, aid with mathematical tractability but are to some extent inessential to the resulting phenomena. For example, while the problem of characterizing the joint distribution of causal variants is more difficult in a population subject to disequilibrium AM, we found (results not shown) that estimators behave in a similar, albeit less extreme, fashion relative to their behavior in an equilibrium population. Other assumptions, such as the absence of gene-environment correlation (which may occur in the presence of vertical transmission) and the conditional independence of mates' genotypes given their phenotypes (which may be violated in structured populations), are more difficult to evaluate and deserve consideration in future investigations.

Reviewer #3 (Remarks to the Author):

I think this is a timely and important paper and definitely needs to be published. The authors have done such a thorough job that I don't have much to complain about.

1. If I could say anything, I would only say that it would be good if they could give simple explanations for why h_0 differs from h_{inf} and why REML asymptotes to h_{inf} and not h_0 in the main text (as opposed to the methods and Sup Mat).

This point is well taken and we have adjusted the manuscript accordingly. Specifically, we further clarify the distinction between the panmictic and equilibrium heritabilities in the Discussion, writing:

By characterizing the full joint distribution of causal variants at equilibrium, we prove that REML produces a consistent estimator of h_0^2 (the heritability in the ancestral

random-mating population), not h_{∞}^2 (the present generation heritability for a trait at equilibrium), in large samples...

And later, in the “implications” subsection:

Fourth, our results demonstrate that controlling for genomic principal components does not mitigate AM-induced biases; indeed, it is the asymptotic equivalence of the spectral distributions of the eigenvalues of the GRM under random versus assortative mating that causes the REML heritability estimator to approach the panmictic heritability in high dimensions. As AM is in essence a form of population structure, our results reveal that our current understanding of the extent to which residual population structure might impact estimators of quantitative genetic parameters is incomplete.

2. The fact that REML converges to h_0 is interesting because it makes the inherent assumption that the effects are independent, yet independence is not necessary for it to be applicable as what James Lee and I tried to show (Lee and Chow. “Conditions for the Validity of SNP-Based Heritability Estimation.” *Human Genetics* 133, no. 8 (2014): 1011–22. <https://doi.org/10.1007/s00439-014-1441-5>). I wonder if the new machinery developed in the Sup Mat could elucidate those results further.

We agree that conceiving of SNP effects as i.i.d. random variates, while perhaps a useful fiction, is not necessary, and causes problems in the case of AM specifically by constraining the variance of the true polygenic score to be equal to the sum of the variances of the effects of individual loci (thereby disregarding the long-range LD induced by AM). We hope that our results will spur the development of more flexible models, and write in the Discussion:

As such, these results should provide motivation and a starting place for the development of new methods that can provide unbiased estimates of genomic variance in the presence of AM and other factors that can influence the long-range correlations between SNPs.

Reviewer #4 (Remarks to the Author):

Dear editor and authors:

The study by Border et al. addresses important issues related to SNP-heritability estimates, including both the effect of assortative mating (AM, the main focus), and finite-sample biases (and the consistency of) two commonly used estimation methods,

The paper is well written and represents an important contribution. I hope my comments will help the authors to further improve their manuscript.

Major comments

1. Variance components versus ‘heritability’ (I am referring to parameters, not estimates). The derivations presented by the authors can be used to trace the effects of AM on the variance of genetic values in a population. However, establishing the

consequences of AM on heritability is more difficult because AM will also induce a correlation of environmental effects (habits) that will also be passed to progeny, thus inducing a genetic-by-environment correlation. This issue is mentioned (but somehow tangentially) in the discussion. I think that this needs to be further highlighted in the manuscript (including possible abstract and introduction) because this is a very serious concern. (I am personally more interested in variances than variance-ratios!)

There are two important ideas in this comment: the circumstances under which AM induces gene-environment correlations (rGE) and the distinction between variance components and variance-ratios.

With respect to the former, our manuscript employs a genotype-phenotype model that assumes that parental phenotypes are independent of offspring environment and that the variance explained by non-heritable factors is constant across generations. The independence of parent phenotype and offspring environments (hereafter, “P/O independence”) is a strong assumption, which will be appropriate for some traits and inappropriate for others, but providing P/O independence holds, AM will not induce rGE. Under P/O independence, the distinction between variance components and variance-ratios is relatively straightforward: For a trait with unit variance at generation 0, the additive genetic variance $V_G[0]$ and the heritability $h^2[0]$ are equivalent, and at generation $t > 0$, we can express additive genetic variance in terms of the current and generation 0 heritabilities:

$$V_G[t] = h^2[t] (1 - h^2[0]) / (1 - h^2[t])$$

Likewise, our primary theoretical results can also apply to variance components: the REML variance-components will converge in probability to their panmictic counterparts in high dimensions, and corrected MoM estimators of the equilibrium and panmictic additive genetic variance can be derived from those we present for heritability using the definition of the heritability.

On the other hand, as noted in Reviewer #4’s comment, when parental phenotypes influence offspring phenotypes apart from the transmission of genetic material, things can be considerably more complicated. For example, when parents transmit environmental factors to offspring (i.e., under vertical transmission), which is likely to be the case for a number of phenotypes, allele counts at all trait-increasing (resp. -decreasing) loci for the trait under assortment will become positively (resp. negatively) correlated with transmissible trait-increasing environmental factors, thereby both increasing the additive genetic variance as before, but further increasing the total phenotypic variance through gene-environment covariance as well as increasing environmental variance. But this is not the only way the P/O independence assumption can fail. For example, under “active” or “evocative” rGE models, child phenotype influences future environmental exposures, and resulting in positive and/or negative feedback cycles between inherited (parental) and environmental influences on a given trait, even in the absence of vertical transmission. Furthermore, environmental influences that are shared between parent and offspring but not passed down from parent to offspring would create P-O covariance but not lead to a gene-environment covariance.

The consequences of AM for variance component estimators in these various contexts are complex, and though certainly deserving of future study, are beyond the scope of the current

investigation. Still, the point is well-taken and we have revised various passages throughout the manuscript both to make our assumptions clearer and emphasize the resulting limitations and directions for future work. Specifically, we write

- in the Introduction:

We assume Fisher's classical model of AM, which describes the equilibrium properties of a heritable trait for which mates' genotypes are conditionally independent given their phenotypes and for which, that offspring environments are independent of parental phenotypes. This model, and which has formed the theoretical foundation for recent investigations of AM using measured genetic data [4, 14, 22].

- In the Discussion:

Other assumptions, such as the absence of gene-environment correlation (which may occur, for example, in the presence of vertical transmission) and the conditional independence of mates' genotypes given their phenotypes (which may be violated in structured populations), are more difficult to evaluate and deserve consideration in future investigations. For instance, when environmental influences are transmitted from parents to offspring, the dependence between causal variants will extend to non-heritable factors such that, in subsequent generations, all trait-increasing allele-counts and trait-increasing environmental factors will become positively correlated, and the linear mixed model assumed by both the REML and MoM estimators will be further misspecified. ... As such, these results should provide motivation and a starting place for the development of new methods that can provide unbiased estimates of genomic variance in the presence of AM and other factors that can influence the long-range correlations between SNPs.

- Equation (5) in the online methods and the preceding sentence state that non-heritable factors are considered as non-heritable factors, which is later restated in the "simulation framework" subsection:

At each generation, phenotypes were constructed via $y=Zu+e$ where e was i.i.d. Gaussian with zero expectation and variance σ_e^2 .

2. Should we care about panmictic or real-population heritability? I would argue that for most applications we are interested in the proportion of variance of phenotypes explained by genetic values in the population, and not on that proportion in (a conceptual) panmictic population. This is addressed in the study, which derives bias relative to h_{∞}^2 and h_0^2 . Perhaps this distinction may need to be further strengthened in the discussion.

We have added language to the discussion to emphasize this distinction, writing:

By characterizing the full joint distribution of causal variants at equilibrium, we prove that REML produces a consistent estimator of h_0^2 (the heritability in the ancestral random-mating population), not h_{∞}^2 (the present generation heritability for a trait at equilibrium), in large samples (see Supp. Materials S2, S3).

Note: In the manuscript, you present a result showing that the REML estimate converges to the heritability in the panmictic population; however, this assumes exchangeability which may not hold under some circumstances (e.g., selection).

We note this limitation in the Discussion, writing:

Finally, we have not investigated the impact that natural selection has on the long-range correlations between causal variants and SNPs and what impact this might have on h^2_{SNP} .

However, it is worth mentioning that we make the exchangeability assumption primarily to aid in tractability but should not be strictly necessary provided the trait is sufficiently polygenic: exchangeability is not used in our characterization of the equilibrium distribution of causal variants, the matrix equations that characterize the behavior of MoM estimators can be reformulated as integrals with respect to an arbitrary distribution on the causal variant effects, and the methods used to characterize the high-dimensional behavior of the REML estimator are compatible with alternative effect distributions provided the conditional distribution of the effects given the genotypes are known, which we discuss in the Supplemental Methods S3.2.1.

3. The demonstration that REML estimates can have a sizable finite sample bias is extremely significant. The same finding was reported four years ago by Mandal et al. (JASA, 2017, see link below) using data from UK-Biobank and human height.

We are grateful to the reviewer for bringing this work to our attention. We cite this previous result in the Results section, writing:

With respect to height specifically, our results are congruent with those of Mandal and colleagues, who independently observed that h^2_{REML} decreased with increasing sample size [26].

4. BIAS of REML estimates. When the amount of information about variance parameters is limited, the REML functions are typically flat around the maximum (which leads to SE) and, importantly, asymmetric (with a long right tail) which sets the conditions for the estimator to be upwardly biased. As the sample size increases, the likelihood becomes sharper and increasingly symmetric, and the bias decreases. In trying to dig into possible factors that may affect bias, I have observed that the shape of the likelihood (or REML) function depends on the % of variance explained by SNPs, sample size, and the number of SNPs. You state that the bias of REML estimates is not affected by the number of SNPs. I am not convinced that this is the case. I've been seriously concerned about the upward bias of estimates derived with ultra-high-density SNPs (e.g., imputed genotypes). Many papers using ultra-high-density SNPs claim that the missing heritability is no longer there; I have not been convinced that is the case.

This is a valid concern, one which we share, and reveals an important point of clarification to be added to the manuscript. Specifically, we meant to imply that, in line with Jiang and colleagues for the REML case (Annals of Statistics, 2016), neither the total number of variants p nor the ratio of causal to non-causal variants m/p impacts the high dimensional behavior of either the REML or MoM estimator, provided that the trait is highly polygenic and that individual variant effects are $O(m^{-1/2})$ in magnitude. Basically, it's okay if the phenotype has a non-infinitesimal architecture as long as there are many causal variants and they all have small effects. However,

this really applies to the hypothetical case where there is no population structure (apart from AM) and that the set of variants under consideration doesn't change qualitatively as its cardinality increases. In the real world, as we study larger numbers of variants, we are likely considering increasingly rare and/or uncertain variants (corresponding to decreasing MAF or imputation quality thresholds, or perhaps as we go from array to WGS data). If the variants being added to the model are capturing more and more fine-scale residual population structure, this will inflate genetic variance component estimates and could lead to severe upward biases. This is something that is critical to address, but also something our results don't speak to, as they assume all causal variants are independent of environmental factors. We have revised the manuscript to clarify this distinction, writing in the Methods:

There were no apparent differences across simulations as a function of m , p , or N_{sim} (Supp. Figure S1). We note that these simulations assume a homogenous population without any population structure apart from that induced by primary phenotypic AM; thus, the irrelevance of p here does not necessarily speak to the consequences of increasing the number of variants under consideration by decreasing allele frequency or imputation quality score thresholds, as is common practice, if the additional variants capture additional fine-scale population structure.

Further, we have amended our discussion of WGS vs. array data in the Discussion to emphasize this point:

Therefore, the difference in REML estimates from a smaller sample of whole-genome sequence data (which ostensibly includes previously missing causal variants) versus those derived from a larger sample of SNP array data will appear artificially greater, though we note that additional factors, such as differences in the extents to which array versus sequence data are corrupted by subtle forms of population structure, might also contribute to such discrepancies. Fourth, our results demonstrate that the controlling for genomic principal components does not mitigate AM-induced biases; indeed, the asymptotic distributions of the eigenvalues of the GRM under random versus assortative mating are equivalent. As AM is in essence a form of population structure, our results reveal that our current understanding of the extent to which residual population structure might impact estimators of quantitative genetic parameters is incomplete.

Other important comments

5. The fact that LD contributes to variance and that this is not fully accounted for in traditional models (which assume iid SNP effects) have been discussed in de los Campos et al. (PLoS Genetics, 2015). I believe that reference is highly relevant for your work.

We address this comment and comment (8) together. We agree that de los Campos et al. 2015 is quite relevant and now cite it in the manuscript, noting that the long-range LD induced by AM further complicates the relationship between variance component estimates and the population genetic parameters of interest. Further, we agree that our implication (that the biases induced by AM are equally relevant when causal variants are genotyped compared with an imperfect LD situation) was unwarranted. We no longer make this assertion and identify the relationship between heritability estimates and the degree of missing heritability under AM as a target for future work. In the Results, we write:

In real-world applications, measured genotypes will often include only a fraction of the total number of causal variants (or at least good proxies in high LD). Previous work has demonstrated that the extent to which marker data imperfectly tag causal variants will bias h^2_{REML} estimates in random mating populations [de Los Campos et al, 2015].; these circumstances are further complicated under AM as imperfect tagging will not only corrupt the relationship between a given marker and proximal causal variants, but between a given marker and all other causal variants. To assess the impact of AM when not all variants are measured, we compared h^2_{HE} and h^2_{REML} in simulated data that randomly dropped = 0%, 50%, or 75% of all $p=10^6$ variants, including both causal ($m=10^4$) and non-causal ($p-m=990,000$) SNPs (Figure 3b). As expected, this resulted in attenuated estimates commensurate with, but not proportional to the fraction of missing data: on average across methods, h^2_{SNP} was 8.4% ($se=0.009\%$) and 17.6% ($se=0.009\%$) lower after randomly dropping 50% and 75% of SNPs, respectively (the degree of attenuation was smaller than the proportion of SNPs dropped due to LD between retained SNPs and discarded causal variants). Though future work is required to fully characterize the relationship between heritability estimates and the degree of missing heritability under AM, the pattern of bias due to AM when some of the heritability is missing is qualitatively similar to that when all causal variants are present.

6. Your work may also have implications for estimates of genetic (or SNP-correlations) as imperfect LD between markers and QTL, and LD between loci can have consequences on covariances as well. the consequences of LD (and imperfect LD) on genetic (and SNP-based) covariances are more difficult to predict than the same consequences on variances (e.g., Gianola et al., GENETICS, 2018). What would happen for instance when both traits are subject to AM in both (or opposite) directions? These issues may be worth discussing as well.

We whole-heartedly agree that the impact of AM on genetic correlation estimates is relevant and are currently pursuing this avenue of research. However, as the multivariate case is considerably more elaborate, we believe that a more thorough treatment is necessary than what would fall within the current manuscript's scope and space constraints. We point to examining genetic correlation as a target for future research in the discussion, writing

Finally, we have not considered the effects of multivariate AM or the impact of AM on marker-based genetic correlation estimators, which will also be misspecified under cross-trait AM. As such, these results should provide motivation and a starting place for the development of new methods that can provide unbiased estimates of genomic variance in the presence of AM and other factors that can influence the long-range correlations between SNPs.

7. "Finally, we demonstrate that these biases affect real traits subject to assortment." I don't think the empirical results demonstrate the biases (because the true parameter values are unknown). Consider re-writing.

We agree that the original language was too strong and have revised the statement to reflect that the empirical behavior of these estimators are consistent with what we'd expect under AM. We now write:

Finally, we demonstrate that the empirical patterns of estimates across methods and

sample sizes for real traits subject to AM are congruent with expected AM-induced biases.

8. The conclusions derived about biases when not all causal variants are genotyped (page 6) are not very convincing to me. We see a drop in SNP-heritability estimates, but that does not lead to concluding that biases are equally relevant when causal variants are genotyped compared with an imperfect LD situation. Consider re-writing or elaborating a bit more.

See our response to point (5) above.

REVIEWER COMMENTS

Reviewer #3 (Remarks to the Author):

I am satisfied with the revisions and recommend publication.

Reviewer #4 (Remarks to the Author):

The authors have addressed all my comments in the revision. Both the revision and the response to reviewers are satisfactory.

I have no further comments.